# Phase-Locked Synthetic Wavelength Interferometer Using a Femtosecond Laser for Absolute Distance Measurement without Cyclic Error

**DOI:** 10.3390/s23146253

**Published:** 2023-07-09

**Authors:** Hyeokin Kang, Joohyung Lee, Young-Jin Kim, Seung-Woo Kim

**Affiliations:** 1Department of Mechanical System Design Engineering, Seoul National University of Science and Technology (Seoultech), 232 Gongneung-ro, Nowon-gu, Seoul 01811, Republic of Korea; 2Department of Mechanical Engineering, Korea Advanced Institute of Science and Technology (KAIST), Science Town, Daejeon 34141, Republic of Korea

**Keywords:** synthetic wavelength interferometer, femtosecond laser, absolute distance measurement, cyclic error

## Abstract

We present a phase-locked synthetic wavelength interferometer that enables a complete elimination of cyclic errors in absolute distance measurements. With this method, the phase difference between the reference and measurement paths is fed back into a phase lock-in system, which is then used to control the synthetic wavelength and set the phase difference to zero using an external cavity acousto-optic modulator. We validated the cyclic error removal of the proposed phase-locked method by comparing it with the conventional phase-measuring method of the synthetic wavelength interferometer. By analyzing the locked error signal, we achieved a precision of 0.6 mrad in phase without any observed cyclic errors.

## 1. Introduction

The unprecedented optical frequency stability and ultrashort pulse width of femtosecond lasers greatly improves the precision and long-range measurement capabilities of conventional interferometers [1,2,3,4,5,6,7]. In the frequency domain, the utilization of optical frequency combs, which consist of well-defined and evenly spaced optical modes, allows for the extension of the non-ambiguity range of conventional interferometers while preserving the high precision achieved by phase-measuring interferometers. These comb-based interferometers leverage the high coherence properties of optical frequency combs to reference the optical frequency of an external tunable laser to the comb, or directly extract and amplify specific optical comb modes [8,9,10]. The use of femtosecond laser-based dispersive interferometers also enables the measurement of long distances, which was previously limited by the visibility degradation of conventional broadband light sources. By utilizing the narrow linewidth and coherence properties of femtosecond lasers, long-distance measurements can be achieved in the spectrally resolved interferometers [11,12,13]. In addition to improving the performance of conventional interferometers, femtosecond lasers have also enabled the existence of multiple heterodyne beats between a number of stable optical modes, such as dual-comb interferometers, which were not available using traditional lasers [14,15]. The dual-comb based absolute distance measurement has been made possible by analyzing the individual phase information of each optical mode in the frequency domain, or by measuring the timing difference of multiple interference signals in the time domain [16,17,18,19,20]. Additionally, ultrafast time-resolving power attributed to the ultrashort pulse and high-peak power has surpassed the precision of conventional time-of-flight distance measurements, which were limited by the bandwidth of photodetection or time-counting electronics [21,22].

Among the aforementioned attempts, synthetic wavelength interferometry (SWI) is a principle that exploits multiple synthetic waves generated by multiple beats between optical modes and has powerful advantages, including the ability to be implemented using a relatively simple apparatus [7,23,24]. The capability of the SWIs to perform long-distance measurements is enabled by using a synthetic wavelength that is longer than the length of the object being measured. In addition, the precision of the SWIs is achieved by using a shortest synthetic wavelength, which can be provided by high-bandwidth photodetection and a superheterodyne-aided phase-measuring technique [25].

As with conventional phase-measuring interferometers, the SWIs also suffer from cyclic errors that can affect the accuracy and precision of the measurements [26,27]. The cyclic errors of interferometers originate from internal multiple path, polarization mixing, RF-frequencies cross-talk, and other factors [26,27,28,29,30,31]. Accordingly, efforts have been made to mitigate the error by optical power stabilization, photonic microwave mixing, data post-processing, iterative compensation, and polarization control [32,33,34,35,36,37].

In this study, we applied a phase-locking technique to the SWI for the complete elimination of the cyclic error. This method involves feeding the phase difference between the reference and measurement path back to a phase lock-in system, which is then used to control the synthetic wavelength and set the phase difference to zero. We used an external cavity acousto-optic modulator (AOM) to generate additional sub-mode synthetic wavelengths, resulting from multiple beats between the original optical modes and the shifted optical modes generated by the AOM. The sub-mode synthetic wavelength provided by the AOM offers a wider tunable range and higher bandwidth than those of conventional intracavity repetition rate control methods for the phase-locked SWI. The performance of the phase-locked SWI was evaluated by measuring the distance calibrated using a commercial heterodyne laser interferometer. To verify the elimination of cyclic errors, we compared the results of the phase-locked SWI with those of conventional phase-measuring SWI.

## 2. Basic Principle and System Implementation

Figure 1 shows the system configuration of the phase-locked synthetic wavelength interferometer (SWI) using a femtosecond laser in this investigation. This system consists of an SWI based on a femtosecond laser and electronics including superheterodyne detection and phase-locked loop (PLL). An Er-doped fiber femtosecond laser (house-built) was used as the light source, providing a pulse train of 100-fs duration with a 100-MHz repetition rate (*f_r_*) stabilized to the Rb clock with a 10^−12^ stability (10-s averaging). The input light source was divided into two beams using the 2 × 1 fiber coupler of a Mach–Zehnder interferometer (MZI). Each beam was then directed into separate fiber arms containing a fiber-coupled AOM (G&H, T-M040), and a fiber polarization controller (Thorlabs, FPC561), respectively. The output of the AOM was designed to allow for only the first-order frequency shift, resulting in a frequency-shifted optical comb as a modulation frequency (*f_m_*) range of 40 MHz ± 6 MHz. A polarization controller was used to manually adjust the polarization state of the light to maximize the beat signals. The two arms were then recombined using a second 2 × 1 fiber coupler. The result was the RF-comb, which consists of fundamental modes *mf_r_*, and AOM-generated sub-modes *mf_r_* ± *f_m_*, where *m* is integer, as shown in the inset diagram in Figure 1.

The output from the MZI was collimated by a lens and then passed through a linear polarizer and a half-wave plate to adjust power ratio between reference and measurement beams. The beam reflected from the PBS was directly incident on a photodetector (Thorlabs, FPD510) installed on the reference arm. The other beam transmitted at the PBS propagated along the measurement arm, including a quarter wave plate, and was detected using the other photodetector, after being reflected back from a retro-reflector.

The retro-reflector was placed on a motorized stage intended to change the measurement distance, which was calibrated using a commercial heterodyne interferometer (Agilent 5517C). To minimize the cosine error between the SWI and the heterodyne interferometer, the propagation paths of the two laser beams were overlapped using a dichroic mirror. The frequencies of the two detected signals were down-converted to the kHz range using the superheterodyne detection circuit and the modulation frequency was fed back to the AOM to set the phase difference between the two signals as zero.

The detailed signal processing and feedback control electronics are described in Figure 2. The two signals from the PDs are filtered using two bandpass filters (BPF, Minicircuits, ZABP-450-S+), allowing only the desired synthetic frequencies to be transmitted. Amplifiers (AMP, Minicircuits, ZFL-500+) were used to improve the signal-to-noise ratio of the corresponding signals. The electronics also included a local oscillator and double-balanced mixers (DBM, Minicircuits, ZAD-1+) for down-converting the signal to the kHz range while maintaining phase, and low-pass filters (LPF, Minicircuits, BLP-5+) to suppress any unnecessary high frequency generated in the mixing process. We added manual phase shifter (PS, Minicircuits, JSPHS-446+) in the reference signal path to adjust the phase-locking point. Additionally, a splitter was applied to the measurement signal path to monitor the frequency of the synthetic wave using a frequency counter (FC, KEYSIGHT, 53220A). The phase error signal, which is generated by a mixer in the phase-locked loop, is loop-filtered and fed-back to a proportional-integral servo (Newport, LB1005) with a 1 kHz control bandwidth. The servo then controls the modulation frequency of the AOM to lock the phase difference into zero. For the purpose of this demonstration, we utilized table-top active electronics including servo, FC and LO, and passive RF components connected with coaxial RF cables. However, for robust and reliable long-term operation, it is desirable to implement an opto-electronic control board with surface-mounted RF components, and monolithic electronics that can function equivalently [38]. Such an implementation would ensure stability, durability, and ease of integration into practical applications.

Similar to conventional phase-measuring homodyne or heterodyne interferometers, SWIs also suffer from cyclic errors resulting from the mixing of the original signal and cross-talk signal generated by signal beam multipath, polarization mixing, and RF signal mixing. As shown in Figure 3a, the cyclic error can be modeled by the phasor representation of the ideal and the cross-talk signal [26,32,37]. The measured signal can be described as the sum of the ideal and cross-talk signal, which can be expressed as follows:(1)$$\begin{array}{cc} A_{m}\cos\left(2\pi f_{syn}t+\theta_{m}\right) & =A_{i}\cos\left(2\pi f_{syn}t+\theta_{i}\right)+A_{c}\cos\left(2\pi f_{syn}t+\theta_{c}\right)\\ & =A_{i}[\cos\left(2\pi f_{syn}t+\theta_{i}\right)+\beta \cos\left(2\pi f_{syn}t+\theta_{c}\right)] \end{array}$$
where *A_m_*, *A_i_*, and *A_c_* are the amplitudes of the measured signal, ideal signal, and cross-talk signal, respectively. *f_syn_* is the frequency of the selected synthetic wavelength generated by the superheterodyne detection electronics, *θ_m_* is phase of the measured signal with the cyclic error, *θ_i_* is phase of the ideal signal, *θ_c_* is the phase of the cross-talk signal, and *β* is the amplitude ratio between the cross-talk signal and the ideal signal. Then, the measured phase, *θ_m_*, can be determined as in the following equation.
(2)$$\theta_{m}=\theta_{i}+{tan}^{-1}\left(\frac{\beta sin(\theta_{c}-\theta_{i})}{1+\beta sin(\theta_{c}-\theta_{i})}\right)$$

As a result, instead of measuring the phase of ideal signal, *θ_i_*, *θ_m_* is measured, leading to periodic distance errors, as shown in Figure 3b. The cyclic error depends on the distance being measured, and increases with the amplitude of the cross-talk signal. As *β* increases from 0.01 to 0.1, the cyclic error also increases from 0.3% to 3% in the peak-to-peak value. However, at the phase position of *mπ*, the cyclic error can be completely eliminated. Therefore, we chose the zero-phase as the phase-locking position to eliminate the cyclic error by controlling the frequency of the synthetic wavelength. Therefore, the distance *D* is determined via integer multiples of synthetic wavelength as follows:(3)$$D=m\frac{c}{{2f}_{syn}N}=m\frac{c}{2(nf_{r}+f_{m})N}$$
where *m* is a multiple integer, *c* is the speed of light, *N* is the group refractive index of air, *f_r_* is the repetition rate of the femtosecond laser, *n* is the multiple integer of the repetition rate, *f_m_* is the modulation frequency of the AOM. The integer number *m* can be determined using a stroboscopic technique [17,21]. The modulation frequency of the AOM is detuned from *f_m_*_1_ to *f_m_*_2_, which are successive frequencies satisfying the phase-locking state. *m* can be calculated as *m* = (*nf_r_* + *f_m_*_2_*)/(f_m_*_2_
*− f_m_*_1_*)*.

**Figure 3 sensors-23-06253-f003:**
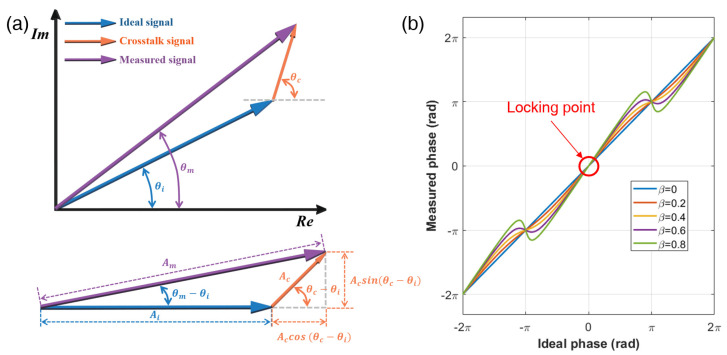
The cyclic error of phase measuring interferometers caused by the signal mixing of the ideal signal with the cross-talk signal. (**a**) Phasor representation of the signal mixing. *A*_i_, *A*_c_, and *A*_m_ (*θ*_i_, *θ*_c_, and *θ*_m)_ are the amplitudes (phase) of the ideal signal, cross-talk signal, and mixed signal, respectively. The presence of cross-talk signal leads to signal mixing, resulting in the measurement of a mixed signal instead of the ideal signal. (**b**) Measured phase as function of the phase of the ideal signal under the presence of cross-talk signal exist. The phase of the measured signal shows cyclic error with comparison to ideal phase. The zero-phase is the phase-locking position to eliminate the cyclic error by controlling the frequency of the synthetic wavelength. β: amplitude ratio between the cross-talk signal and the ideal signal.

In absolute distance measurement, it is often challenging to simultaneously achieve both high precision and a large measurement range. This trade-off between precision and range can indeed be a key consideration in various industrial and scientific applications. The precision of the phase-locked SWI can be evaluated using Equation (4), which is derived from differentiating Equation (3).
(4)$$\delta D=m\left|\frac{c}{{{2f}_{syn}}^{2}N}\right|\delta f_{syn}=m\left|\frac{c}{{{2(nf}_{r}+f_{m})}^{2}N}\right|(n\delta f_{r}+\delta f_{m})$$

The precision, represented by *δD*, is primarily influenced by the phase-locked frequency of the synthetic wavelength and its associated noise. To enhance precision, it is beneficial to use a higher frequency for the synthetic wavelength, similar to the conventional phase-measuring SWI. This can be achieved by increasing the multiple integer, *n*, of the repetition rate, which can be controlled by tuning the local oscillator (LO) frequency in the superheterodyne circuit, as depicted in Figure 2. On the other hand, it is important to consider the impact of phase-locked frequency noise, denoted as *δf_syn_*, on the overall precision. This noise consists of two main components: the noise related to the repetition rate and the performance of the phase-locked loop (PLL) circuit. As the measurement distance increases, the multiple integer, *m*, in Equation (4) for precision increases. This leads to an amplification effect of the parameters influencing precision, which are dependent on the distance. Therefore, there is a trade-off between the measurement distance and the precision that needs to be carefully adjusted based on the specific requirements and applications of absolute distance measurement.

Owing to the limited control range of the synthetic wave, there exist dead zones where the phase-locking is not available. The position that requires the maximum frequency control occurs when the phase difference between the reference and measurement signals is π, as shown in Figure 4a. The distance can then be described as a multiple of the synthetic wavelength, along with a fraction of π, as shown in the following equation:(5)$$\frac{2ND}{c}=m\frac{1}{f_{syn1}}-\frac{1}{{2f}_{syn1}}$$
where *f_syn_*_1_ denotes the frequency of the synthetic wavelength before phase-locking, and 1/2*f_syn_*_1_ denotes the fraction of π. As shown in Figure 4b, a frequency detuning from *f_syn_*_1_ to *f_syn_*_2_ occurs, resulting in the phase difference between the reference and measurement signals being in-phase, allowing for phase-locking. The distance can be described using the following equation: (6)$$\frac{2ND}{c}=m\frac{1}{f_{syn2}}$$

Using Equations (5) and (6), the required frequency detuning at distance D is reached using the following equation: (7)$$\left|f_{syn2}-f_{syn1}\right|=\delta f_{syn}=\frac{c}{4ND}$$

The dead zone problem is exacerbated at short distances because a large frequency shift is required to achieve a zero-phase difference between the reference and the measurement signals, as shown in Figure 4c. The red circle in the figure represents the control range of the AOM used in this study, which is 40 MHz ± 6 MHz. It indicates that there is no dead zone occurring beyond 6.25 m, as the AOM’s control range is sufficient to cover the required frequency shift for phase-locking at those distances.

**Figure 4 sensors-23-06253-f004:**
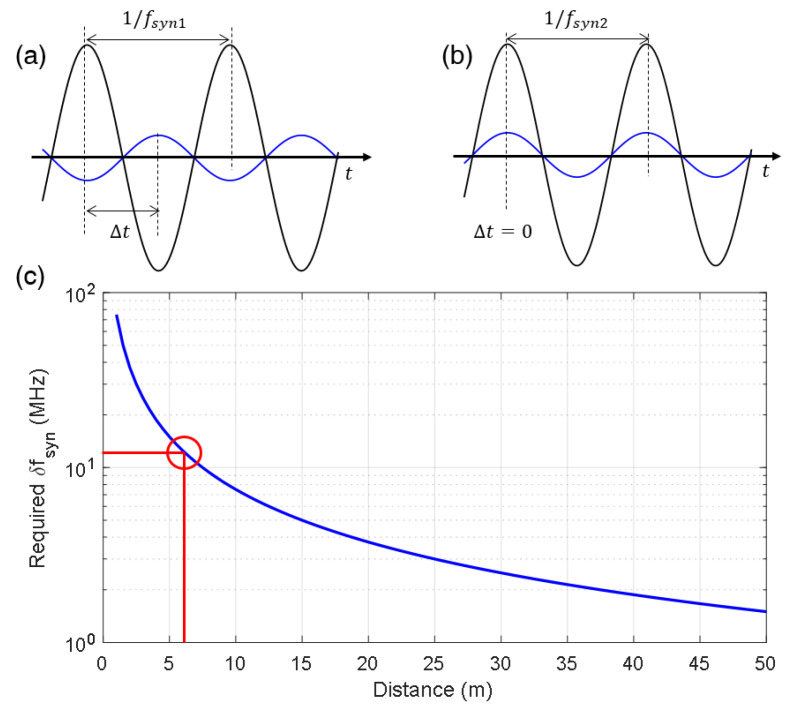
Dead zone problem of the phase-locked SWI. (**a**) Maximum frequency control is required for the phase locking, when phase difference between the reference and measurement signals is *π*. Blue and black line denotes reference signal and measurement signal, respectively. (**b**) Two signals are in-phase after the frequency is detuned from *f_syn_*_1_ to *f_syn_*_2_. (**c**) Maximum required frequency for dead zone-free distance measurement depending on distance to be measured. The red circle indicates that there is no dead zone occurring after 6.25 m due to the 12 MHz control range of the AOM in this study.

## 3. Results

Figure 5a shows the successful generation of the RF-comb in the MZI that incorporates the AOM, with a modulation frequency set to 40 MHz. The RF-comb was measured using an RF-spectrum analyzer (R&S, FSIQ 7) and 200 MHz in −3 dB bandwidth photodetector (Thorlabs, FPD510). The measured RF-comb contains multiples of 100 MHz corresponding to the repetition rate of the femtosecond laser used, as well as sub-peaks induced by the AOM at $mf_{r}\pm 40\;MHz$. Additional sub-peaks were observed at $mf_{r}\pm 20\;MHz$, presumably generated by the coupling of second-order beams in addition to the first-order beam in the fiber coupled AOM used. In this experiment, we selected the 440 MHz peak, corresponding to a synthetic wavelength of 0.682 m. The linewidth of the peak was measured to be 0.6 Hz in full width at half maximum (FWHM), as shown in Figure 5b. This narrow linewidth indicates that there is no degradation in linewidth compared to the fundamental harmonics of the laser, and that it is suitable for long-distance measurements.

After evaluation of the selected synthetic wavelength, we demonstrated a distance measurement using the phase-locked SWI without cyclic error, as shown in Figure 6. The retro-reflector on the motorized stage was translated into 25 steps over 480 mm range. To ensure that the cyclic error occurred for at least one full period, we added an ~18.7 m fiber spool (SMF-28) into the measurement arm. This allowed us to measure distances beyond the dead zone limit, which was within the control range of the AOM. The integer number *m* was determined to be 164, accounting the optical path difference between the reference arm and the fiber-added measurement arm. Furthermore, to demonstrate the effectiveness of cyclic error removal by the phase-locking, we intentionally detuned the QWP in measurement arm, resulting in an increase of cyclic error due to the cross-talk mixing caused by the multipath of the measurement beam. During the translation, the AOM continuously locked the phase at zero by controlling the modulation frequency from 41.3 MHz to 37.5 MHz, which was measured by the frequency counter at 1 s averaging time converting to distance using Equation (3). The distance was simultaneously measured using a commercial heterodyne interferometer with a nanometer-level precision. After the phase-locked SWI measurement, we repeated the measurement without phase-locking by measuring the phase of the synthetic wave using a phasemeter (Stanford Research, SR865) to monitor the cyclic error. Figure 6a shows the measurement result of the phase-locked SWI and the conventional phase-measuring SWI. To compare the errors of each SWI, we subtracted the linear term from the results, as shown in Figure 6b. The phase-measuring SWI clearly showed a cyclic error of 0.41 rad in the peak-to-peak value, corresponding to a cross-talk signal mixing ratio of ~0.2, calculated using Equation (2). The severe cross-talk error was clearly removed by the phase-locked measurement, which had a peak-to-peak error of less than 0.05 rad, as shown in Figure 6b. The observed peak-to-peak error of the phase-locked SWI can be attributed to the use of a long fiber considering the precision analysis that will be conducted in the following paragraph.

Figure 7a shows the error signal obtained from the servo in the phase-locked SWI for 25 stepped positions over 480 mm. Each individual step is depicted with a sequential black and gray color. We used oscilloscope (Agilent, 54832B) to sample the error signal with setting 10 kHz sampling rate, corresponding to averaging time of 10 μs for total of 25 s measurement time. During the measurement, the error signals at each position were successfully maintained at the zero phase without phase-locking expiration. To evaluate the precision of the phase-locked SWI, the Allan deviation was calculated for both the stepwise and fixed position, as shown in Figure 7b. For the stepwise movement case, the data from Figure 7a was used directly. For a fixed position, the distance was maintained at the center position of the stage, and the locked signal was recorded using the identical sampling method as in the stepwise movement case. The Allan deviation of the fixed position exhibited slightly better performance than that of the stepwise movement for 10^−4^ to 1 s averaging time. However, it is worth noting that the entire range of values was within the error bars, indicating that the phase-locking of the SWI was not significantly degraded in the stepwise movement case. This can be attributed to the high control bandwidth of the AOM, which ensures stable and precise phase-locking throughout the measurement. The Allan deviation of the phase-locked SWI at 1 s averaging time was estimated to be 0.6 mrad, which corresponds to a precision of 32.6 μm considering the synthetic wavelength of 0.682 m used. Once the frequency of the synthetic wavelength is increased up to ~15 GHz, which is readily possible with commercial products, sub-μm precision in absolute distance measurement is achievable without the cyclic error.

## 4. Conclusions

We demonstrated cyclic-error-free absolute distance measurement using the phase-locked synthetic wavelength interferometer with an external cavity AOM. To overcome the cyclic error caused by various origins of signal mixing, we calculated the phase-locking point at which the cyclic error was eliminated through simulations, and then locked the phase at that point during the measurement. We then performed distance measuring using the 440 MHz AOM-induced peak with the phase-locking and conventional phase-measuring method, sequentially. By comparing the results with a commercial laser interferometer, we demonstrated that the cyclic error can be completely removed. The high-bandwidth of the AOM resulted in tight phase-locking during the measurement, and narrow linewidth of synthetic wave, which enabled the capability of measuring long distance. By analyzing the error signal from the servo, the phase-locking SWI showed precision of 32.6 µm. Once the frequency of the synthetic wavelength is increased up to ~15 GHz, sub-µm precision in absolute distance measurement will be available without the cyclic error. The cyclic-error-free measurement will enable accurate and reliable distance measurements, particularly in industrial and scientific applications that require high-precision and long-distance measurements.

## Figures and Tables

**Figure 1 sensors-23-06253-f001:**
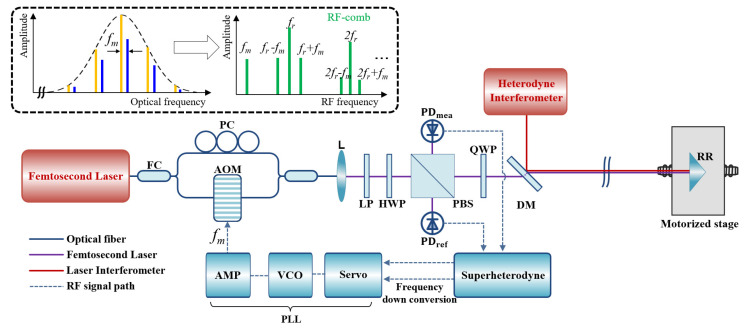
System configuration of the phase-locked synthetic wavelength interferometer using a femtosecond laser. (inset) RF-comb generation by multiple beats between original optical comb (yellow) and *f_m_* shifted optical comb (blue) by an AOM incorporated fiber Mach–Zehnder interferometer. To evaluate the performance of the distance measurement, a motorized stage is utilized in the measurement arm to vary the distance between the SWI and a retro-reflector. A commercial heterodyne interferometer is adopted to validate the distance measurement result obtained from the phase-locked SWI. The control signal for the phase-locking between two RF signals is generated using a superheterodyne detection circuit and a phase-locked loop circuit including a servo, a VCO, and an RF amplifier. The resulting control signal is then fed back to the acousto-optic modulator. Abbreviations are; FC: fiber coupler, PC: polarization controller, AOM: acousto-optic modulator, L: collimating lens, LP: linear polarizer, HWP: half-wave plate, PBS: polarizing beam splitter, QWP: quarter-wave plate, DM: dichroic mirror, PD_ref(mea)_: photodetector for reference (measurement) signal, RR: retro-reflector, PLL: phase-locked loop, VCO: voltage-controlled oscillator, *f_r_*: pulse repetition rate, *f_m_*: modulation frequency.

**Figure 2 sensors-23-06253-f002:**
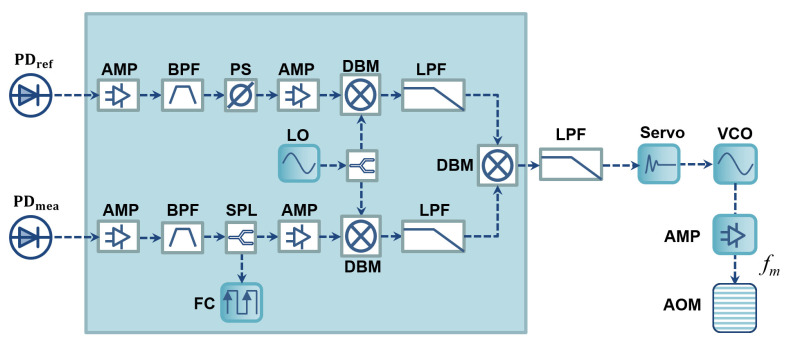
RF electronics for the phase-locked SWI. The control signal generated by the two RF signals from the photodetectors is processed through a superheterodyne circuit and a phase-locked loop circuit. The resulting signal is then fed back to the acousto-optic modulator (AOM) through a VCO and an RF-amplifier. Abbreviations: AMP: RF-amplifier, BPF: bandpass filter; PS: phase shifter; DBM: double balanced mixer; LPF: lowpass filter; VCO: voltage controlled oscillator; SPL: splitter; FC: Frequency counter; LO: Local oscillator.

**Figure 5 sensors-23-06253-f005:**
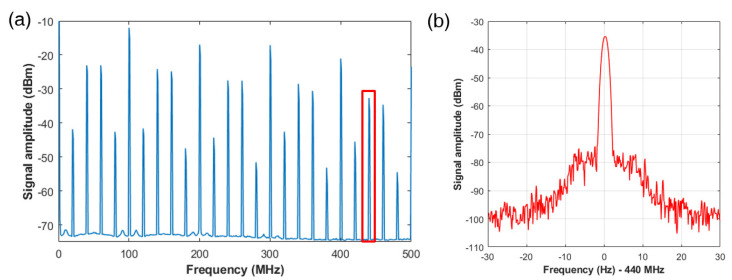
The RF-comb generation from the fiber MZI incorporating with the AOM. (**a**) Spectrum of RF-comb includes peaks of the harmonics of repetition rate of the fiber laser, and sub-peaks induced by the AOM. The repetition rate of the fiber laser was stabilized at 100 MHz, and the modulation frequency of the AOM was set to 40 MHz. The red box indicates the selected synthetic wavelength with a frequency of 440 MHz, which is used for distance measurement in the phase-locked SWI. (RBW: 50 kHz, VBW: 100 Hz) (**b**) Linewidth measurement of the 440 MHz frequency (RBW: 1 Hz, VBW: 1 Hz). The linewidth of the peak was measured to be 0.6 Hz in full width at half maximum (FWHM).

**Figure 6 sensors-23-06253-f006:**
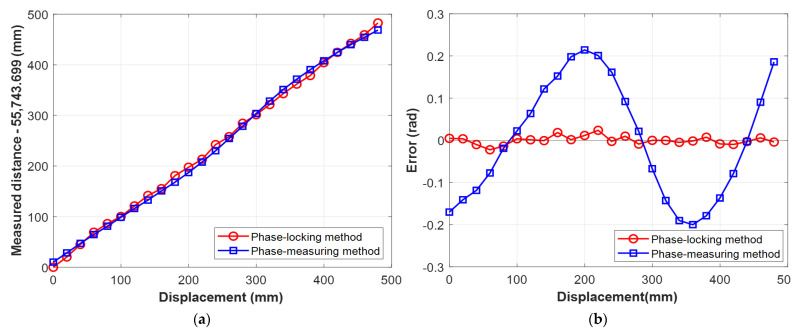
(**a**) Measurement result of the phase-locked SWI and conventional phase-measuring SWI, increasing the distance using a motorized stage. The displacement was calibrated with the commercial heterodyne interferometer. (**b**) Error of the two SWIs with respect to the commercial heterodyne interferometer. (Blue) Due to the signal cross-talk caused by beam multipath in the measurement arm, the cyclic error is clearly shown in the phase-measuring SWI. (Red) The cyclic error is removed in the phase-locked SWI.

**Figure 7 sensors-23-06253-f007:**
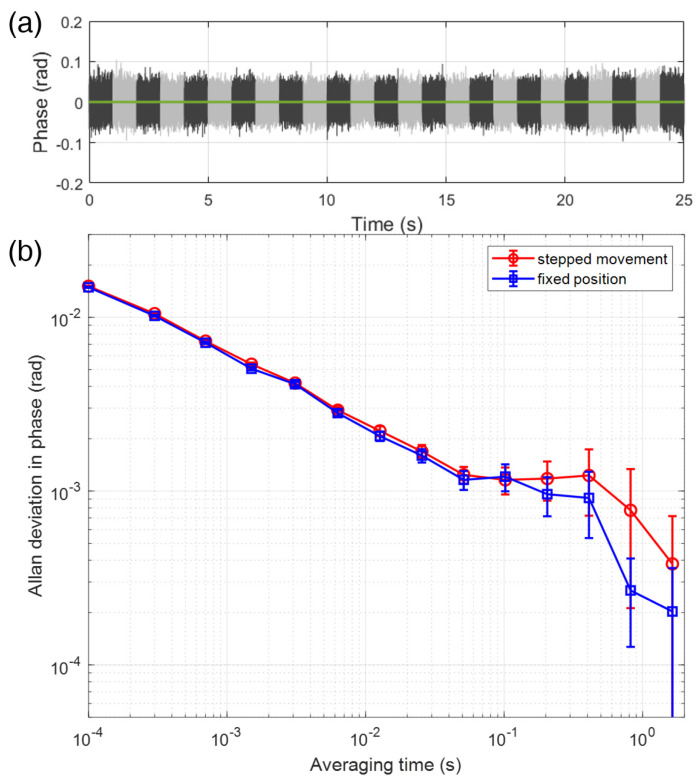
Precision analysis of the phase-locked SWI using the synthetic wavelength of 0.682 m (440 MHz) (**a**) The error signal of the phase-locked SWI during the 25 stepwise movement over 480 mm. The error from the servo was recorded for 25 s with 10 kHz sampling rate. Each individual step is represented by a sequential black and gray color. During the measurement, the error signals at each position were effectively maintained at the zero phase without any phase-locking expiration. (**b**) Allan deviation in phase with respect to averaging time for both the stepped position (red) and fixed position (blue) of the phase-locked SWI. The Allan deviation of the phase-locked SWI at a 1 s averaging time was determined to be 0.6 mrad, which translates to a precision of 32.6 µm considering the synthetic wavelength of 0.682 m that was used in the measurement. The Allan deviation values were observed to be within the error bars, indicating that the phase-locking of the SWI was not significantly affected in the case of the stepwise movement. This robust performance can be attributed to the high control bandwidth of the AOM, which enables stable and precise phase-locking throughout the measurement process.

## Data Availability

Not applicable.

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
