# Peer review of "Phase-Locked Synthetic Wavelength Interferometer Using a Femtosecond Laser for Absolute Distance Measurement without Cyclic Error"

_sensors, 2023, doi:10.3390/s23146253_

Round 1
Reviewer 1 Report
The authors demonstrated a method to construct a cyclic-error-free absolute distance measurement system using the phase-locked synthetic wavelength interferometer with an AO modulator. The method was proved effective at the synthetic wavelength of 0.682 m and it was predicted sub-μm precision is feasible if the synthetic is increased with commercial products. The method is interesting, practical and should be helpful for absolute distance measurement with rf combs. The manuscript is organized well and the illustration of the technological details is clear. The following issues need to be improved.
1) In absolute distance measurement, usually high precision and large measurement range can not be achieved simultaneously. The authors should discuss the measurement range, which can be a key issue in industrial and scientific applications.
2) In Fig. 3 (b), the caption should be corrected because the horizontal axis is not ‘distance’.
3) What about the robustness of the phase-locked SWI? Although acceptable, the feedback electronics is quite complicated. I am not sure if it can be stably locked at the zero-phase-difference state in long-time operation.
4) Too many self-citations. At least 14 out of 42 references are published by the same group.
Author Response
We sincerely appreciate the valuable remarks provided by the reviewers, demonstrating their deep understanding of our work. We have carefully considered each remark and have made corresponding revisions in the revised manuscript. In the attached file, you will find a point-by-point response document where we address each remark and describe the changes we have made.

Reviewer 2 Report
The paper describes an interesting system for removing the cyclic-error in phase measuring interferometer based on synthetic wavelength generation.
The paper is well written, and I believe it deserves publication. However, in my opinion, there are some aspects which should be clarified of better explained. Hereafter, list of major and minor comments following the order of the paper.
1) At line 37 at page 1 “to improving” must be replace with “to improve”
2) In Figure 1 the PLL block should be clearly highlighted for better comprehension.
3) The acronym NPR at page 3 must be extended.
4) In Eq. 3) I think there should be or a “+” or a “-“ sign at the denominator. Otherwise two possible distances come out. Moreover, the following paragraph regarding the tuning of the frequency fm must be explained more clearly, maybe with some more computations if necessary.
5) In Eqs. (4) and (5) I think that the refractive index N should be at the numerator, right? Then the first two lines of paragraph just after Eq. (4) (up to “…group refractive index of air” are just a repetition of what has been mentioned after Eq. (3) and in my opinion can be eliminated.
6) The word “is” at line 185 must be removed.
7) In Section 3 it is said that a 200MHz PD has been used, but the spectrum shown goes to 500MHz. Maybe it should be specified that 200MHz is the 3dB BW.
8) In figure 6a, what does the number expressed on the y-axis represent? An offset?
9) Can you give a comment on the behavior of the error of figure 6b with respect to the displacement? For example what is the reason why this error occur and why we have such regular behavior.
As final remark, I think the work is very good and the system proposed shows very interesting results. However, In general, I recommend more detailed explanations, as in the list above, to improve the quality and the clearness of the work.
Author Response

(The authors gave the same response as above.)

Round 2
Reviewer 2 Report
In my opinion the quality of the paper has been improve sufficiently for publication.